# OPENCRONIC Study. Knowledge and Experiences of Spanish Patients and Carers about Chronic Disease

**DOI:** 10.3390/ijerph16010039

**Published:** 2018-12-24

**Authors:** Emilio Casariego, Ana M. Cebrián-Cuenca, José Luis Llisterri, Rafael Manuel Micó-Pérez, Domingo Orozco-Beltran, Mercedes Otero-Cacabelos, Pilar Román-Sánchez, Francisco José Sáez

**Affiliations:** 1Internal Medicine Department, Hospital Universitario Lucus Augusti, 27003 Lugo, Spain; emilio.casariego.vales@sergas.es; 2Cartagena Casco Health Centre, Cartagena, 30201 Murcia, Spain; anicebrian@gmail.com; 3Ingeniero Joaquin Benlloch Health Centre, 46006 Valencia, Spain; jllisterric@gmail.com; 4Ontinyent Primary Care Team, Fontanars dels Alforins Clinic, 46635 Valencia, Spain; rafaelmmicoperez@gmail.com; 5Research Unit, Alicante-Sant Joan D’Alacant Department, Cabo Huertas Health Centre, 03540 Alicante, Spain; 6Marazuela Clinic Emergency Department, Talavera de la Reina, 45600 Toledo, Spain; moc@semg.es; 7Internal Medicine Department, Hospital General de Requena, Requena, 46340 Valencia, Spain; roman_pil@gva.es; 8Primary Care Management Department of the Madrid Health Service, 28035 Madrid, Spain; fsaezm@semg.es

**Keywords:** knowledge, chronic disease, serious adverse events, carer, survey

## Abstract

***Background***: Chronic diseases are currently the main cause of morbidity and mortality and represent a major challenge to healthcare systems. The objective of this study is to know Spanish public opinion about chronic disease and how it affects their daily lives. ***Methods***: Through a telephone or online survey of 24 questions, data was gathered on the characteristics of the respondents and their knowledge and experiences of chronic diseases. ***Results***: Of the 2522 survey respondents, 325 had a chronic disease and were carers, 1088 had a chronic disease and were not carers, 140 did not have a chronic disease but were carers, and 969 did not have chronic disease and were not carers. The degree of knowledge on these diseases was good or very good for 69.4%, 56.0%, 62.2%, and 46.7%, respectively, for each group. All the groups agreed that chronic diseases mainly affect mood, quality of life and having to make sacrifices. ***Conclusions***: Knowledge about chronic diseases is relatively good, although it can be improved among the Spanish population, especially among patients who report having a chronic disease and play the role of carers. However, it is important to continue maintaining the level of information and training concerning these diseases.

## 1. Introduction

Until the last decades of the 20th century, infectious diseases were considered the main cause of morbidity and mortality. However, since then the pattern has changed, giving way to the prevalence of chronic diseases. According to the World Health Organisation (WHO), chronic diseases are defined as long-term illnesses, and in general, progress slowly [1]. Even so, the definition may vary in the scientific literature and from one healthcare organisation to another, with this variation even extending to the list of diseases included in this category [2]. The WHO considers heart disease, heart attacks, cancer, respiratory disease, and diabetes to be the most significant chronic diseases, and the main causes of mortality in the world, responsible for 63% of deaths.

In Spain, it has been calculated that chronic diseases are the cause of 92% of the total number of deaths, with the majority of these being cardiovascular diseases (31%) and cancers (28%). The probability of dying between the ages of 30 and 70 due to the four main chronic diseases (cancer, diabetes, cardiovascular disease, and chronic respiratory disease) is 11% [3]. As well as being one of the main causes of death, chronic diseases also represent one of the main reasons for dependence and loss of quality of life by patients [4,5]. The changeable and behavioural risk factors for the contraction of chronic diseases are the consumption of tobacco, physical inactivity, dangerous consumption of alcohol, and an unhealthy diet [6]. According to a survey carried out by the Ministry of Health in Spain between 2011 and 2012, 75.3% of the population see their personal health as good or very good. However, one in six adults suffer from some of the most frequent chronic disorders: Lower back pain (18.6%), high blood pressure (18.5%), arthrosis, arthritis or rheumatism (18.3%), high cholesterol (16.4%), and chronic neck pain (15.9); all of these display an upward trend and are more common in women. Furthermore, 53% of the population over 65 do not have any functional dependency problems. This same survey assessed health-related quality of life in adults for the first time, outlining that the issue that most frequently causes problems is pain or discomfort, which affects 24.8% of the population aged 15 years or older. This is followed by anxiety and depression, which affects 14.6%, and problems walking, which affects 13.9% [7].

These diseases pose a significant challenge for healthcare systems, especially in terms of their sustainability, which is something that has already been confirmed by the Organisation for Economic Co-operation and Development (OECD) [8,9]. Therefore, measures need to be set out to increase patient quality of life and to reduce the care and economic burden involved. Prevention, diagnosis, management, and monitoring of chronic diseases constitutes the foundation of the care model for chronic patients, and it is internationally accepted as the main strategic response to the challenges of chronic diseases [10]. This model requires reassessment of the care for these patients to provide permanent, coordinated, and multidisciplinary systems [11]. The model consists of the following main elements: Professionals and care processes that provide proactive care; planning and coordination of care; visit and follow-up programmes; support in the decision-making process, including guidelines and protocols for the management of diseases based on the best available evidence; information systems to guarantee access to appropriate and relevant information; support for patient empowerment and self-care; community resources to inform and support patients; and support for the chronic disease care system by the professionals that make up the healthcare network [12].

With the aim of being able to establish measures to improve patient care and lower complications in the treatment of these diseases, it is important to take the perspectives of patients and their carers into account. However, until now no study evaluating this factor has been carried out among the Spanish population. It is in this context that the OPENCRONIC study (Estudio de OPinión de la población española sobre la ENfermedad CRÓNICa [Spanish population opinion poll on chronic diseases]) was developed. Its objective was to gather the opinions of both patients and the general Spanish population about the knowledge of certain chronic diseases, how they affect their daily lives, and the role of carers in their management.

## 2. Methods

### 2.1. Study Design

A cross-sectional, stratified, observational study carried out between October and November 2017 with a representative sample of the Spanish population of 40 years or older through a system of surveys developed and conducted by the market research consulting firm, Hamilton. Before their inclusion, all participants were informed about the study and gave their consent to participate in it.

### 2.2. Sample Selection

The study consists of surveys carried out on individuals of 40 years or older, from both people not having a chronic diseases (general population) and those that have some type of chronic disease (chronic patients). Chronic patients were considered to be those that informed us that they were affected by one or more of the diseases stipulated by the study coordinators, these being: Alzheimer’s, senile dementia, Parkinson’s, arthritis or another rheumatoid disease, arthrosis, asthma, high cholesterol levels, depression or anxiety, type 1 or 2 diabetes, chronic obstructive pulmonary disease (COPD), non-valvular atrial fibrillation, hepatitis B, high blood pressure, heart failure, chronic liver failure, and chronic kidney failure. Diagnosis could not be confirmed in any of the cases. Among those surveyed with chronic diseases, clinical events are considered to be hospital visits due to severe events such as decompensation, myocardial infarction, stroke, hypoglycaemia, respiratory failure, etc.

In both groups (general population and chronic patients), the subject may or may not have been acting as a carer. Carers were considered to be anyone that was dedicating at least 8 h a week to the care of a chronic disease patient.

The selection of the number of respondents was not probabilistic, but was made through quotas that were proportional to the actual Spanish population in terms of age, sex, and level of education, according to data provided by the Instituto Nacional de Estadística [Spanish National Institute of Statistics] (INE). Slight deviations between theoretical and actual distribution in the sample were corrected through weighting coefficients. We started with a sample containing 2522 valid surveyed subjects with a margin of error of 1.99%, a confidence interval of 95.5% and a maximum indeterminacy of *p* = *q* = 50. Of these, 1022 were surveyed by telephone and 1500 were surveyed online. To arrive at these figures, 42,851 phone calls were made and 15,000 online invitations were sent.

### 2.3. Procedure

A mixed system of telephone and online surveys was used with the idea that mixing both procedures together would ensure greater coverage of all sociodemographic profiles, thereby compensating for the underrepresentation of young people with a landline and older people with internet access.

A telephone survey of approximately 20 min was carried out through the computer-assisted telephone interviewing (CATI) system, which randomly selects telephone numbers. Calls that were able to complete the survey were considered valid. At the same time, online surveys were conducted using the computer-assisted web interviewing (CAWI) methodology via prior dispatch of invitations with distribution proportional to the established quotas based on Spain’s resident population. Sex, age, and autonomous community quotas were established, calculated by taking into account internet penetration according to data obtained through the INE [13].

Both types of survey were distributed according to age and educational level, and deviations were corrected through weighting of the data using age and educational level data provided by the INE.

### 2.4. Questionnaire

Both the telephone and the online questionnaire were identical. They consisted of 24 questions, of which 4 were of a general nature about sociodemographic characteristics and 20 about the subjects’ knowledge and experience of chronic diseases. For each question, several possible responses were included for the respondent to choose from. The questions were proposed by Hamilton and validated by a group of experts in chronic diseases in a face-to-face meeting.

### 2.5. Data Analysis

This is a study aimed at a centred population with a quantitative base. The statistical analysis was carried out on overall results that were broken down according to different variables of interest to bring varied insights on the knowledge, perception, and attitudes of the Spanish population regarding chronic diseases.

The results were first broken down according to whether the respondent had a chronic disease (chronic patients) or belonged to the group that did not have a chronic disease (general population). Each group of subjects was then divided according to whether or not they were a carer for a chronic disease patient.

The data analysis was descriptive, using frequencies and percentages for the categorical variables and via the mean and standard deviation (normal variables) or the median, range, and Q1 and Q3 quartiles (variables without normal distribution) for continuous variables. The categorical variables were compared using the exact Fisher test and the continuous variables were compared using the *T*-test or the Mann-Whitney U test. The statistical significance was set at *p* < 0.05. The program used for the data analysis process was IBM SPSS^®^ Statistics (IBM Corp. Released 2015. IBM SPSS Statistics for Windows, Version 23.0. Armonk, NY, USA).

### 2.6. Data Confidentiality

The survey was conducted according to current legislation regarding data protection (Directive 95/46/EC, Organic Law 15/1999 of 13 December on personal data protection and additional regulations). The data were gathered anonymously.

## 3. Results

Of the 2522 respondents, 1413 (56.0%) stated that they have a chronic disease and 1109 (44.0%) were part of the general population without chronic diseases. Within the group with chronic diseases, 325 (12.9%) stated that they were carers and 1088 (43.1%) were not carers. For the population group who did not have chronic diseases, 140 (5.6%) were carers and 969 (38.4%) were not carers (Table 1).

The distribution of respondents by age matches that described by the INE, with a greater percentage of those aged 70 years or over (20.5%) [14]. Within the group of respondents with chronic diseases, the majority only had one disease (54.6%), which was painless (61.1%) and without the need for hospitalisation (74.7%), but which was being treated using medicinal products (86.7%). Of the chronic diseases described, the most frequent were high blood pressure (21.5%) and arthritis/arthrosis (18.9%). The rest were mood conditions, such as depression or anxiety (10.4%), type 1 or 2 diabetes (9.8%), respiratory diseases, such as COPD or asthma (6.0%), and atrial fibrillation or heart failure (3.5%) (Table 1).

As regards the level of information that the respondents had on chronic diseases, out of the total number of people surveyed, 54.5% considered themselves to be quite or very well-informed (12.6% very well-informed and 41.9% quite well-informed). The level of information was higher when the respondent was a carer, regardless of whether they themselves had a chronic disease (69.4% quite or very well-informed) or were free from chronic disease (62.2% quite or very well-informed). Only those respondents who were not carers and did not have a chronic disease showed lower levels of knowledge (46.7% quite or very well-informed) (Figure 1).

Overall, the respondents consider mood, having to make sacrifices, the loss of quality of life, or having to make radical lifestyle changes to be the aspects most affected due to having a chronic disease (55.5%, 55.3%, 53.1%, and 53.0%, respectively) (Table 2). For patients with a chronic disease who are carers, the aspect most affected by these diseases is having to make sacrifices (58.5%) and the aspect that is least affected is social relationships (31.0%). For patients with a chronic disease who are not carers, the main aspect affected is a loss in quality of life (46.8%) and the aspect that is least affected is family relationships (23.3%). For carers from the general population, the aspect most affected is mood (71.5%) and that which is least affected is family relationships. Finally, for non-carers from the general population, the aspect most affected by chronic diseases is mood (64.2%) and the least affected is social relationships (Table 2).

Most notably, the implications of having a chronic disease with which the respondents agreed strongly or quite strongly included: The increased risk of serious events (heart attacks, strokes, etc.), becoming dependent at a younger age, and having to make radical lifestyle changes. However, not all the respondents showed the same level of knowledge about these. In all cases, those who were most well-informed were patients with a chronic disease (63.6%, 64.6%, and 61.7% with a high level of information for each implication, respectively) (Table 3).

## 4. Discussion

The survey carried out in the present study on the knowledge and experiences of patients, carers, and the general population has given an insight into the opinions held by Spanish people regarding chronic diseases and how they affect daily life. Although other surveys on knowledge of these diseases have already been conducted, most of them have only focused on a single disease [15,16,17]. In 1988, Corbin and Strauss published a work in which they addressed some of the problems of chronic disease management at home, describing how patients and their partners manage chronic illness and at the same time how they deal with the effects of the disease on their lives [18]. Since then, this is the first time a study has been published on chronic diseases in general, with special reference to the role of carers.

The degree of knowledge about any disease, and chronic diseases in particular, is of vital importance for their effective management, especially in terms of self-management by patients [10]. It has been established that a low level of knowledge of the disease produces worse results, higher hospitalisation rates, increased use of emergency services, worse adherence to treatment due to not knowing how to take medication, and higher mortality rates [19]. On the other hand, carrying out educational and training initiatives increases the degree of knowledge on the disease and reduces many of the problems associated with it [20]. However, due to the fact that healthcare systems are designed to deal mainly with acute illnesses, the effort required to deal with chronic diseases is often not made, and patient training and level of knowledge is neglected [10].

In spite of all this, the degree of knowledge held by the Spanish population regarding chronic diseases is not particularly low. Of those surveyed, 54.5% stated that they were very or quite well informed regarding chronic diseases, 33.0% that they were somewhat informed and only 12.4% stated that they were not very or not at all informed. In a similar survey carried out in Australia regarding patients’ level of knowledge on diseases such as diabetes, hyperlipidemia, and cardiovascular diseases, the percentage of patients with no knowledge of the disease was 5%, 27%, and 15% respectively [17]. Within Spain, ignorance regarding COPD ranges from between 91% among people in Valencia to 73.7% among people in Aragon [15]. At the other end of the spectrum is osteoporosis, with a degree of knowledge among Spanish women of 82.5% [16]. An additional aspect of our study looked at the level of information among people in the role of carer, regardless of whether or not they have a chronic disease, which was seen to be 69.4% and 62.2%, respectively. In this regard, as described by a systematic review of interventions with the carers of chronic disease patients, the more information they receive, the higher the quality of life [21].

While for people with chronic diseases, regardless of whether or not they are carers, the aspects considered to be most affected by chronic diseases involve having to make sacrifices and quality of life, for those who do not have a chronic disease, the aspects most affected are loss of quality of life and mood. On the other hand, all of them considered that social or family relationships are the aspects that are least affected. It has been described that one of the main changes generated by chronic diseases, and perhaps the one that causes the greatest deterioration in patients, is the emotional burden, as subjects are forced to adapt their lifestyle in the long-term, generating a series of normally negative emotions such as fear, anger, and anxiety [22]. Some diseases, such as cancer, not only provoke changes in the patient due to the adaptation process involved, they can also lead to financial difficulties, change the person’s self-perception and affect their relationships with family and friends [22]. These same aspects have also been described for patients with chronic kidney failure [23]. At the same time, 88% of the care required by chronic patients is offered by social support networks, especially the closest family members [24].

The limitations of this study are the same as those normally seen in survey-based studies, such as not being able to confirm the diagnosis of the diseases described and that the perceptions of the chronic diseases are subjective. In spite of its limitations, this type of survey is widely used by the Spanish National Health Survey and other similar surveys, both nationally and internationally. While some people responded to the survey online and others by telephone, we do not consider that this has conditioned the responses, because although those who respond via the online system would normally have more time than those who answer the telephone survey, the telephone respondents have the opportunity to clarify any issues regarding the interpretation of the questions with the interviewer. Another limitation of the study is that data were not gathered from patients with chronic diseases that are prevalent in subjects under 40 years old. However, it is important to bear in mind that most chronic disease patients are older than this. The study’s main strength is that the deviations in the results were corrected through weighting of the data using age and educational level data provided by the INE.

## 5. Conclusions

The level of knowledge regarding chronic diseases among all the people surveyed was relatively good, especially among those having such illnesses who also act as carers. However, this level could still be improved upon, even more so if we bear in mind that over 80% of the people surveyed did not act as carers and had a lower level of knowledge. Therefore, it is important to continue highlighting the role of education and training among the population regarding these diseases. Thanks to the results obtained, measures can be established that take into account the perspectives of patients and their carers and which reduce subsequent complications in their treatment. Given the possible cultural and social differences between countries, and even between their health systems, similar studies are needed to understand each country’s chronic disease situation.

## Figures and Tables

**Figure 1 ijerph-16-00039-f001:**
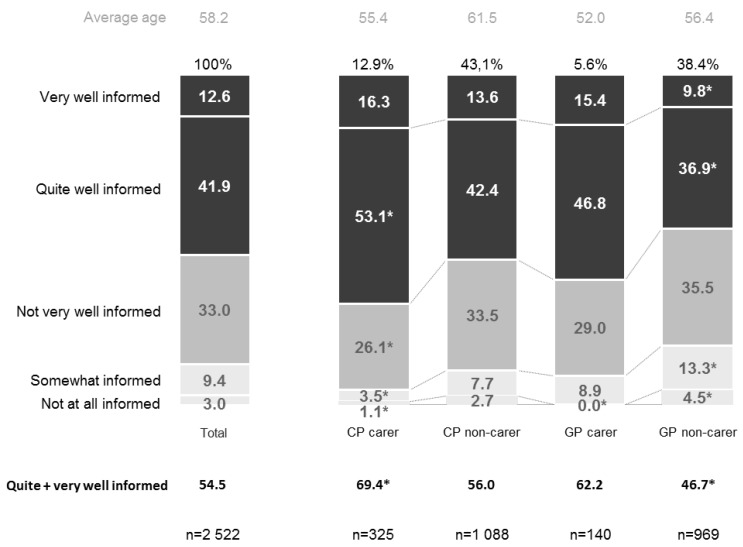
Level of information of the respondents regarding chronic diseases. CP: Chronic patient; GP: General population; * *p* < 0.05 (comparisons with respect to the total).

**Table 1 ijerph-16-00039-t001:** Characteristics of the respondents (*n* = 2522).

Characteristics	*N*	%
**Age**		
40–44 years	347	13.8
45–49 years	329	13.0
50–54 years	321	12.7
55–59 years	342	13.6
60–64 years	335	13.3
65–69 years	331	13.1
70 years or over	517	20.5
**Educational Level**		
Elementary/no studies	414	16.4
Primary/secondary education	1154	45.8
Higher education	954	37.8
**General Population**	1109	44.0
Carers of chronic patients	140	12.6 ^#^
Non-carers of chronic patients	969	87.4 ^#^
**Chronic Patients**	1413	55.0
Carers of other chronic patients	325	23.0 ^#^
Non-carers of other chronic patients	1088	76.0 ^#^
With 1 disease	772	54.6 ^#^
With 2 diseases	370	26.2 ^#^
With 3 or more diseases	271	19.2 ^#^
With pain	550	38.9 ^#^
No pain	863	61.1 ^#^
With hospitalisation ^‡^	357	25.3 ^#^
No hospitalisation	1056	74.7 ^#^
With pharmacological treatment	1225	86.7 ^#^
No pharmacological treatment	188	13.3 ^#^
**Diseases Referred to by the Respondents ***		
High blood pressure	542	21.5
Arthritis/arthrosis	477	18.9
Mood disorders (depression/anxiety)	262	10.4
Type 1/2 diabetes	248	9.8
Respiratory diseases (COPD/asthma)	151	6.0
Non-valvular atrial fibrillation/heart failure	89	3.5
Other	212	8.4
**Body Mass Index ****		
General population	1109	100.0
≤24.9 kg/m^2^	542	48.9
25–29.9 kg/m^2^	434	39.1
≥30 kg/m^2^	133	12.0
Chronic patients	1413	100.0
≤24.9 kg/m^2^	506	35.8
25–29.9 kg/m^2^	619	43.8
≥30 kg/m^2^	288	20.4
Total population	2522	100.0
≤24.9 kg/m^2^	1048	41.6
25–29.9 kg/m^2^	1053	41.8
≥30 kg/m^2^	421	16.7

* A respondent could have several diseases. ** Body mass index calculated based on the weight and height indicated by the respondents. ^#^ Percentages over the total general population or chronic disease patients, respectively. ^‡^ Hospitalisation due to severe events, such as decompensation, myocardial infarction, stroke, hypoglycaemia, respiratory failure, etc.

**Table 2 ijerph-16-00039-t002:** Respondents who are affected by the following aspects related to chronic diseases or who believe they must have a significant or quite significant effects.

Having Chronic Diseases Has a Significant or Quite Significant Effect on the Following Aspects	Chronic Patients		General Population		Total
	Carer (*n* = 325)	Non-Carer (*n* = 1088)	Carer (*n* = 140)	Non-Carer (*n* = 969)	
They affect mood	55.5%	45.3% *	71.5% *	64.2% *	55.5%
They involve making sacrifices, for example not being able to consume certain foods or drinks (salt, alcohol, etc.)	58.5%	46.0% *	70.0% *	62.20% *	55.3%
They involve loss of quality of life	51.0%	46.8% *	65.3% *	58.9% *	53.1%
They force you to radically change your lifestyle	54.1%	44.2% *	66.3% *	60.2% *	53.0%
They increase the risk of serious events such as heart attack, stroke, etc.	51.5%	44.0% *	58.4% *	51.7%	48.8%
They increase the risk of early death	47.6%	40.4% *	58.0% *	53.7% *	47.5%
They lead to situations of dependency at a younger age	47.8%	39.6% *	64.9% *	53.8% *	47.5%
They cause professional difficulties/restrictions	44.0%	28.9% *	56.4% *	54.6% *	46.6%
They make you withdrawn and lose the will to do things	45.4%	38.5% *	59.4% *	58.2% *	46.3%
They have a negative effect on social relationships/friendships	31.0%	25.7% *	42.1% *	38.2% *	32.2%
They have a negative effect on family relationships	31.6%	23.3% *	40.7% *	38.5% *	31.2%

* *p* < 0.05 (comparisons with respect to the total).

**Table 3 ijerph-16-00039-t003:** Self-declared level of information of the respondents who agreed strongly or quite strongly with the following implications of having a chronic disease.

Implications of Having a Chronic Disease with Which the Respondents Agree Strongly or Quite Strongly	High Level of Information		Average Level of Information		Low Level of Information	
	Chronic Disease (%)	General Population (%)	Chronic Disease (%)	General Population (%)	Chronic Disease (%)	General Population (%)
They increase the risk of serious events such as heart attack, stroke, etc.	63.6 ^#^	52.8 *	28.5	33.1	7.9	14.1
They lead to situations of dependency at a younger age	64.6 ^#^	53.6 *	25.4	31.3	10.0	15.1
They force you to radically change your lifestyle	61.7 ^#^	53.1 *	29.3	32.6	9.0	14.3

Chronic patients (*n* = 1413); General population (*n* = 1109). ^#^, *p* < 0.05.

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
