# Peer review of "OPENCRONIC Study. Knowledge and Experiences of Spanish Patients and Carers about Chronic Disease"

_ijerph, 2018, doi:10.3390/ijerph16010039_

Reviewer 1 Report

Thank you for the opportunity to review this manuscript.  This paper illuminates an important portion of the public perspective about chronic illness.  Below are some recommendations to strengthen the manuscript.

Abstract/Intro

There should be consistency of terms throughout the manuscript.  For example: “caregiver” vs. “carer”.  It is assumed it is the same but please clarify.

Methods

Has the survey (questions) used by the consulting firm been validated?   Could you provide more information about the survey, origin of items, item/response text?  It may be beneficial since there is noted variability  of the responses for each item, to include a table with the items and responses in their entirety. This will be useful for readers who may want to conduct a similar telephone/online survey in the future.

Results

While this was a random sample, it would help to understand the geographic distribution of the population that was selected (by region?)  Patients living in urban communities vs. rural communities may have varying access to health care services and therefore different perspectives so it would be interesting to hear more about the geographic setting.

Line 170  There is a much discussion about the level of being “well informed”  Could you clarify what they are well informed about in the context of chronic disease?  (e.g., causes, prevention, medications, treatment, plan of care).

Discussion

2nd paragraph.  These statistics should be included in the results section.

It would also be a great contribution to the literature to aggregate the sample’s responses by disease.  This could have great practice and research implications to inform the scientific community about which disease classes the population are more or less informed about.

Table 1/2/3

There was initial confusion regarding the use of “general population” because it implies that you are comparing the sample to Spain’s general population.  I would recommend using “study sample throughout all tables.

In table 1 please clarify the use the symbol “#” after only some of the reported demographic percentages.  Were all demographics not presented as a percentage over the “total population”?

 Author Response to reviewer 2's comments:

Thanks for your comments.

Abstract/Intro

There should be consistency of terms throughout the manuscript.  For example: “caregiver” vs. “carer”.  It is assumed it is the same but please clarify.

To use the same term throughout the manuscript, all terms in the article have been written as “carer”.

Methods

Has the survey (questions) used by the consulting firm been validated?   Could you provide more information about the survey, origin of items, item/response text?  It may be beneficial since there is noted variability  of the responses for each item, to include a table with the items and responses in their entirety. This will be useful for readers who may want to conduct a similar telephone/online survey in the future.

The questions were proposed by the market research consulting firm Hamilton and validated by a group of experts in chronic diseases in a face-to-face meeting. This information has been added to the article.

We decided not to include the questions in a table because their design is a little complex and could confuse the reader. Many of the questions were linked to each other, so depending on the answer given in one of them, the following questions varied.

Results

While this was a random sample, it would help to understand the geographic distribution of the population that was selected (by region?)  Patients living in urban communities vs. rural communities may have varying access to health care services and therefore different perspectives so it would be interesting to hear more about the geographic setting.

We appreciate the reviewer's proposal. However, given the complexity of the Spanish health system, divided into 17 regions (Autonomous Communities) with their own health system, we decided that a description of results by region was very complex and would distract from the main objective. Nevertheless, it would be interesting to carry out a study that exclusively analyzes the different point of view between urban vs. rural communities of patients and carers about chronic diseases.

Line 170  There is a much discussion about the level of being “well informed”  Could you clarify what they are well informed about in the context of chronic disease?  (e.g., causes, prevention, medications, treatment, plan of care).

The level of information reported by respondents (Figure 1) was on general aspects of chronic diseases, but not on any particular aspect. The respondent only had to choose one of the 5 available options: very well informed; quite well informed; not very well informed; somewhat informed; and not at all informed.

Discussion

2nd paragraph.  These statistics should be included in the results section.

According to reviewer suggestion, this paragraph has been eliminated because these results were already in the results section.

It would also be a great contribution to the literature to aggregate the sample’s responses by disease.  This could have great practice and research implications to inform the scientific community about which disease classes the population are more or less informed about.

We appreciate the suggestion of the reviewer and the importance to the scientific community about including responses by disease. However, we decided not to include those results because we would outgrow the length of the article established by the journal. Taking into account that the respondents reported having 6 different chronic diseases, the length of the results would be increased by 6, because we would have to expose the same comments that we have done in a general way for each one of those diseases.

Table 1/2/3

There was initial confusion regarding the use of “general population” because it implies that you are comparing the sample to Spain’s general population.  I would recommend using “study sample throughout all tables.

As we stated in the Methods section, we considered “general population” to the respondents without a chronic disease. The study sample would include the general population and the population with chronic diseases. We have made a little modification in the “Methods” section to be clearer. However, the table style proposed by the journal is quite confusing because it is difficult to separate the results of both groups of respondents.

In table 1 please clarify the use the symbol “#” after only some of the reported demographic percentages.  Were all demographics not presented as a percentage over the “total population”?

Most of the percentages shown in Table 1 are referred to the entire sample (n=2,522). Only the percentages marked with # are referred to the number of respondents without a chronic disease (general population, n=1,109) o with a chronic disease (n=1,413) respectively.

Reviewer 2 Report

This is a well written article which might be strengthened in the following ways.

1.  The word suffering is not well like by English speaking patients. We usually  talk about people with not suffers. Rather than talk about suffering you can talk able problems or concerns. This is a small point but seems to be very important to patients.  Suffering is a pretty strong word in English.

2.  These findings mirror well the findings of Corbin and Strauss many years ago in a stellar qualitative study where they found the three concerns of chronic disease patients for medical management, role management and emotional management. It might be well to offer this insight as their study was done with English speakers in the US.
 Corbin, Juliet M., and Anselm Strauss. Unending work and care: Managing chronic illness at home. Jossey-Bass, 1988

3.   Instead of general population I would say populating without chronic condition who were not carers.  General population is too broad.

4.   I am surprised that hospitalizations for Diabetes were not included

5.  As the sample is described it would be useful to compare the percent in your sample with the prevalence in Spain to validate the generalizability of the sample.

6.  Some description of the questions asked would be helpful in judging the responses

7.   I think an additional conclusion may be that people with chronic conditions in Spain are not much different than people with chronic conditions elsewhere.  This is important as often for psychoeducational interventions, each country and cultural group feels they need to invent programs specific to their populations which are "different".

Author Response to reviewer 2's comments:

Thanks for your comments.

1.  The word suffering is not well like by English speaking patients. We usually talk about people with not suffers. Rather than talk about suffering you can talk able problems or concerns. This is a small point but seems to be very important to patients.  Suffering is a pretty strong word in English.

We appreciate reviews comments and we have made a little changes to the text to avoid the excessive use of the word “suffering”.

2.  These findings mirror well the findings of Corbin and Strauss many years ago in a stellar qualitative study where they found the three concerns of chronic disease patients for medical management, role management and emotional management. It might be well to offer this insight as their study was done with English speakers in the US. Corbin, Juliet M., and Anselm Strauss. Unending work and care: Managing chronic illness at home. Jossey-Bass, 1988

We agree that the study of Corbin and Strauss is very interesting, although it is very old. However, given the relevance of that study, we have made a brief mention of it at the beginning of the discussion, even though journals do not recommend citing books, especially if they are so old.

3.   Instead of general population I would say populating without chronic condition who were not carers.  General population is too broad.

As we stated in the Methods section, we considered “general population” to the respondents without a chronic disease. The study sample would include the general population and the population with chronic diseases. We have made a little modification in the “Methods” section to be clearer. However, the table style proposed by the journal is quite confusing because it is difficult to separate the results of both groups of respondents.

4.   I am surprised that hospitalizations for Diabetes were not included

As written, it seemed that decompensation, myocardial infarction, stroke, hypoglycaemia or respiratory failure were the only clinical events leading to hospitalization. We have made a small modification to the text to be more general in this regard. Now it can be read: “clinical events are considered to be hospital visits due to severe events such as decompensation, myocardial infarction, stroke, hypoglycaemia, respiratory failure, etc.”

5.  As the sample is described it would be useful to compare the percent in your sample with the prevalence in Spain to validate the generalizability of the sample.

According to the methods section, the selection of the number of respondents was made through quotas that were proportional to the actual Spanish population according to data provided by the Spanish National Institute of Statistics (INE). In other words, the population described in the study is a proportional reflection of the current situation in Spain. However, up to now no study has been carried out to compare the results obtained.

6.  Some description of the questions asked would be helpful in judging the responses

The questions were proposed by the market research consulting firm Hamilton and validated by a group of experts in chronic diseases in a face-to-face meeting. This information has been added to the article. However, we decided not to include the questions in a table because their design is a little complex and could confuse the reader. Many of the questions were linked to each other, so depending on the answer given in one of them, the following questions varied.

7.   I think an additional conclusion may be that people with chronic conditions in Spain are not much different than people with chronic conditions elsewhere.  This is important as often for psychoeducational interventions, each country and cultural group feels they need to invent programs specific to their populations which are "different".

We greatly appreciate the suggestion made. However, we believe that there are cultural and social differences between countries, and even between their health systems that may influence the knowledge and experiences of patients about chronic diseases. We have expanded the conclusion to include this information.